# Intravenous Iron for Perioperative Anaemia in Colorectal Cancer Surgery: A Nested Cohort Analysis

**DOI:** 10.3390/cancers17111877

**Published:** 2025-06-03

**Authors:** Dominic Fritche, Frances Wensley, Yanika L. Johnson, Callum Robins, Mai Wakatsuki, Imogen C. Fecher-Jones, Lisa Sheppard, Malcolm A. West, Alice Aarvold, Mark R. Edwards, Michael P. W. Grocott, James Plumb, Denny Z. H. Levett

**Affiliations:** 1Perioperative and Critical Care Theme, NIHR Southampton Biomedical Research Centre, University of Southampton, Southampton SO16 6YD, UK; dominicfritche@googlemail.com (D.F.);; 2Cancer Sciences, Faculty of Medicine, University of Southampton, Southampton SO16 6YD, UK; 3Clinical and Experimental Sciences, Faculty of Medicine, University of Southampton, Southampton SO16 6YD, UK; 4Salisbury NHS Foundation Trust, Salisbury SP2 8BJ, UK; 5Jersey Hospital, Jersey JE1 3QS, UK; 6University Hospital Southampton NHS Foundation Trust, Southampton SO16 6YD, UK

**Keywords:** anaemia, iron deficiency, haemoglobin, colorectal cancer, gastrointestinal surgery, perioperative care, intravenous iron, length of stay, allogenic blood transfusion

## Abstract

Iron deficiency anaemia (IDA) is common among patients with colorectal cancer (CRC), increasing their risk of blood transfusion and prolonged stays in the hospital. This study uses routinely collected data from patients having surgery for CRC at a UK hospital to see if giving intravenous (IV) iron before their operation reduces blood transfusion and length of stay. Our findings show that IV iron reduces the length of stay in the hospital and the need for a blood transfusion, but it does not affect other outcomes, such as emergency admission to intensive care or readmission to the hospital.

## 1. Introduction

At the point of consideration for surgery, iron deficiency anaemia (IDA) is present in between a quarter and a third of the general surgical population [1,2] and is an independent risk factor for post-operative morbidity and mortality [3]. Patients with anaemia are more likely to receive a blood transfusion, to have a post-operative infection, to be admitted as an emergency to the intensive care unit (ICU), to have a prolonged stay in hospital and to die within 90 days of surgery [4,5,6]. Retrospective analyses have reported reduced blood transfusion rates, complications and lengths of stay associated with the use of intravenous (IV) iron to treat anaemia and iron deficiency before surgery, generating the rapid adoption of this approach into clinical practice in the UK. However, multicentre randomised control trials (RCTs) have not consistently reproduced this effect [7], leading to speculation regarding the true efficacy and effectiveness of IV iron prior to surgery. The pathophysiology leading to iron deficiency and anaemia varies between surgical populations, which may partly explain the heterogeneity in results across multicentre RCTs. Consensus guidelines advocate for the use of perioperative IV iron in surgical patients with IDA but state that these recommendations are based largely on data from small observational studies, with the caveat that any analyses were unlikely to be adequately powered to strongly support recommendations for iron therapy [8]. There is an appreciation that further well-defined homogenous studies would be valuable in identifying best practices regarding treatment of perioperative anaemia in the surgical population. However, such studies may be challenging to perform as the correction of anaemia pre-operatively has become standard care in many regions.

IDA is more common in patients with colorectal cancer (CRC) than in the general surgical population, with a reported prevalence as high as 75% [9]. In patients undergoing elective bowel resections for primary colorectal cancer, anaemia and iron deficiency are caused by two pathological processes: IDA secondary to chronic blood loss within the bowel will cause an absolute IDA, while the inflammation associated with cancer will cause an anaemia of chronic inflammation and functional iron deficiency. This pro-inflammatory state leads to a raised hepcidin, which renders oral iron ineffective [10]. IV iron has, therefore, been proposed as an effective way to bypass the hepcidin pathway and boost iron stores in these patients prior to major surgery. As post-operative anaemia is associated with poor outcomes in CRC patients [1,11], the management of anaemia before surgery may have substantial benefits in this cohort. However, a recent systematic review of RCTs of perioperative IV iron treatment for CRC patients [12] identified only five studies comprising less than 500 patients. The meta-analysis of data from the identified studies showed a reduction in blood transfusion rates for patients receiving IV iron compared to either placebo, standard of care, or oral iron. Furthermore, this analysis demonstrated no difference in major surgical complications, the duration of hospital stays, or 30- or 90-day mortality. The study’s authors highlighted the inevitable clinical and statistical heterogeneity and called for further well-defined studies in this population.

Here, we describe the post-operative outcomes associated with treating IDA with IV iron pre-operatively in a secondary analysis of a prospectively collected CRC cohort (nested within the Fit4surgery cohort) at an NHS tertiary care centre. Our primary outcome was the likelihood of discharge throughout the length of stay in the hospital during admission for an index operation. Secondary outcomes included the allogenic blood transfusion rate in the perioperative period (defined as within 30 days of the operation), unplanned ICU admissions, unplanned return to theatre, thirty-day and eight-week readmission rates and ninety-day mortality rates.

## 2. Materials and Methods

### 2.1. Study Design and Data Sources

The perioperative anaemia service (POAS) at the University Hospital Southampton NHS Foundation Trust (UHS) was established in 2016 to investigate and manage anaemia in patients undergoing major surgery. All patients were followed up as part of the Fit4Surgery cohort, a prospective long-term study of all major surgical patients referred to the perioperative medicine service. Routinely collected data, including patient demographics, comorbidities, standard perioperative blood tests, the timing of treatment, timing of surgery, surgical outcomes and transfusion rates, are stored in the Fit4Surgery Research Database (REC 24/EE/0053, updated in 2024 by the East of England—Essex Research Ethics Committee).

We performed a nested cohort analysis of all elective CRC patients undergoing major surgical resections who were referred POAS from 2016 to 2022. The anaemia service was disrupted during the COVID-19 pandemic, reducing referrals over this period. Audit data reveal that approximately 250 major CRC resections are performed on patients at the UHS annually, of whom 35% are anaemic [13]. Patients were included if they were referred to the POAS and underwent major CRC surgery, including right hemicolectomy, left hemicolectomy, anterior resection, subtotal colectomy, total colectomy, proctocolectomy and abdomino-perineal excision of the rectum (APER). Operations not for a CRC, including palliative surgery, patients < 18 years of age, and patients who received IV iron after surgery as their main form of treatment were excluded. Patients referred to the service during this time who were not anaemic or did not receive treatment were included as reference groups.

Haemoglobin was measured at the point of listing for surgery, and anaemic patients were referred to the POAS for assessment and treatment. Anaemia was defined as a haemoglobin (Hb) level less than 130 g/L in both sexes, given intraoperative blood loss is equivalent [14]. Iron deficiency was defined as ferritin at less than 30 ng/mL or transferrin saturation (TSAT) at less than 20% [8]. Hb (g/L), TSAT (%) and ferritin (ng/mL) were measured pre-infusion, and Hb was used to monitor anaemia perioperatively. Iron deficiency anaemia was treated with IV iron in the form of Monofer^®^ ferric derisomaltose at a dose of up to 20 mg/kg (adjusted for obesity or a low BMI) in line with national guidance [15]. This was administered as a single dose, with further doses administered as needed after a minimum of four weeks.

The primary outcome measure was the length of stay, measured as the probability of remaining in the hospital each day post-operatively. Secondary outcomes were the administration of allogenic blood transfusion, unplanned ICU admissions, unplanned return to theatre, thirty-day and eight-week readmission rates and ninety-day mortality.

### 2.2. Statistical Analysis

Analyses were conducted across three patient groups: patients with anaemia treated with IV iron (treated anaemia), patients with anaemia who were not treated with IV iron (untreated anaemia) and patients without anaemia (non-anaemic). Data were summarised using means and standard deviations (SDs), medians and interquartile ranges (IQRs) or percentages as appropriate. Cox proportional hazard models were used to quantify the effect of anaemia treatment on length of stay. Kaplan–Meier curves were used to compare the likelihood of a patient being in hospital at a given time post-operatively across groups. Models were progressively adjusted for age and sex, comorbidities and surgical details. Secondary outcomes were evaluated using logistic regression models adjusted for age and sex. All analyses and data visualisation were performed in StataNow/SE 18.5.

## 3. Results

Data were available on the 407 patients who were referred to the POAS prior to undergoing major CRC surgery at the UHS from 2016 to 2022. Over this period, a total of 2141 patients underwent colorectal resections at the UHS [13]. This reflects the referral patterns from the surgical teams. Non-anaemic patients are not routinely referred to this service. Of those patients referred to the POAS, 99 were not anaemic, 220 had anaemia and received IV iron treatment (the treated anaemia group), and 88 had anaemia but did not receive IV iron treatment (the untreated anaemia group). The main reasons given for not receiving IV iron were insufficient time between referral to the POAS and surgery (*n* = 44) and historical treatment cutoffs (i.e., Hb level > 120 g/L and <130 g/L were classified as not anaemic at the start of the POAS when the capacity for delivering IV iron therapy was limited (*n* = 39)). One patient declined treatment, one patient was unable to complete their infusion due to a hypersensitivity reaction, and three patients had no recorded reason for not receiving treatment.

The baseline characteristics of the three groups are presented in Table 1. Patients with anaemia were older (71 [SD 12]) vs. (65, [SD 12], *t*-test *p*-value < 0.001) and more likely to be female (52% vs. 29%, Chi^2^
*p*-value < 0.001) than patients without anaemia. The most common surgical procedures were anterior resection and right hemicolectomy, and over two-thirds of all operations were performed laparoscopically. There was no difference in the procedure or surgical approach between anaemic and non-anaemic patient groups (Chi^2^
*p*-value 0.014 and 0.203, respectively). Hb levels throughout admission follow an expected pattern (Figure 1). Pre-operative Hb was 144 g/L (SD 9) in the non-anaemic group, 119 g/L (SD 9) in the treated anaemia group and 119 g/L (SD 12) in the untreated anaemia group (*p*-value < 0.001 for *t*-test comparing patients with and without anaemia).

### 3.1. Treatment with Intravenous Iron

All patients in the treatment group received IV iron in the form of Monofer^®^ as per national guidance [15]. There were two adverse drug reactions (<1% of patients) in the form of pruritis and a new rash, both of which resolved, and the patients completed their infusions without further complications. The median increase in haemoglobin levels pre-operatively was 13 g/L (interquartile range, IQR 4, 23 g/L). The median time from iron infusion to surgery was 25 days (IQR 11, 41). Table 2 shows the mean Hb and TSAT levels, median ferritin levels before IV iron infusion and the mean Hb levels perioperatively across groups.

### 3.2. Clinical Outcomes

Patients with untreated anaemia had a longer length of stay (9 days, IQR 7, 13 days) compared to the other two groups: patients with treated anaemia (7 days, IQR 5, 10) and those who were not anaemic (7 days, IQR 6, 10). Cox proportional hazard models are presented using patients with untreated anaemia as a reference group (Table 3). The hazard ratio (HR) represents the likelihood of being in the hospital on a given day in each group compared to the reference group. For patients in the treatment group, the age- and sex-adjusted HR was 0.60 (95% confidence interval [CI] 0.46–0.77); for patients without anaemia, it was 0.67 (0.48–0.92). Adjusting for additional risk factors such as comorbidities or type of surgery did not change the effect estimates (Table 3). Patients with anaemia who did not receive treatment pre-operatively had an increased likelihood of remaining in hospital on any given day (Figure 2, *p*-value Log-rank test < 0.001).

Mortality rates were low in this population. Thirty-day mortality was 0.74% (*n* = 3), ninety-day mortality was 1.2% (*n* = 5), and six-month mortality was 2.2% (*n* = 9). Deaths were distributed evenly across the three groups (Chi^2^
*p*-value = 0.219). Odds ratios comparing treated anaemia and non-anaemic groups with the untreated anaemia group for secondary clinical outcomes are presented in Table 4. In total, 163 (40%) patients had a planned admission post-operatively to a surgical high-dependency unit and 36 (9%) patients were admitted to intensive care (ICU) post-operatively. In total, 17 (47%) ICU admissions were unplanned or emergency admissions. No difference was detected in readmission rates, unplanned ICU admissions or reoperation rates between the three groups (Table 4).

### 3.3. Transfusion Outcomes

Overall, 13% (*n* = 54) of patients received a blood transfusion perioperatively. Patients with untreated anaemia were more likely to receive a blood transfusion (*n* = 23, 26%), followed by patients with treated anaemia (*n* = 25, 11%). Patients in the non-anaemic group were significantly less likely to receive a blood transfusion in the perioperative period (*n* = 6, 6%, Chi^2^
*p*-value < 0.001). The age- and sex-adjusted odds ratios comparing groups are presented in Table 4. The odds ratio for a perioperative blood transfusion was 0.35 (95% CI 0.18, 0.66) for patients with treated anaemia pre-operatively compared to those with untreated anaemia. Similarly, comparing non-anaemic patients to those with untreated anaemia, the odds ratio was 0.20 (0.07, 0.55). The median number of units transfused in the untreated anaemia group was two (IQR 2, 3) units, compared to one (IQR 1, 2) unit in the treated anaemia group (Wilcoxon–Mann–Whitney test *p*-value 0.030). In non-anaemic patients, the median number of units transfused was two (IQR 2, 2, *p*-value 0.708 compared to the untreated anaemia group).

## 4. Discussion

This single-centre nested cohort study using prospectively collected data provides insights into clinically relevant post-operative outcomes in patients who are treated for IDA in a large tertiary care centre in the UK. Overall, there was a significant reduction in the median length of stay in patients who underwent pre-operative IV iron treatment compared to patients who did not receive treatment for their anaemia. There was also a significant reduction in the blood transfusion rate in patients receiving IV iron. Although patients with treated anaemia received fewer median units of blood (one versus two) compared to the other groups, this difference was not statistically significant and is unlikely to be clinically meaningful. To our knowledge, this is the first reported comparison of the number of units of blood transfused in anaemic and non-anaemic patient groups undergoing major surgery [12]. We found no difference in readmission, reoperation, and unplanned ICU admission rates.

This work responds to recent consensus guidelines calling for further well-defined studies aimed at understanding the relationship between perioperative anaemia and post-operative outcomes [8]. The mean Hb increase in this group was 14 g/L, which is similar to other studies [16], confirming that patients referred to the POAS are adequately dosed with IV iron prior to their operation. A recent retrospective study by Kangasponta et al. [17] demonstrated a reduction in post-operative complications but no difference in blood transfusion rates or length of stay. This cohort was slightly smaller (318 patients) and excluded patients without anaemia.

A recent meta-analysis of the RCTs of colorectal patients who received IV iron prior to operation comprised less than 500 patients across five studies [12]. To the best of our knowledge, this is the only meta-analysis and, therefore, the highest quality of evidence in this field. Four of the five studies in the meta-analysis were reported as low risk of bias, and one study had some concerns about deviation from intervention. Of note, these studies extend back to 2009, all administered different formulations of IV iron, and median time to surgery was slightly lower than recommended by current guidelines. The meta-analysis, which is in keeping with our study, reported reduced rates of perioperative blood transfusion in the treated anaemia groups and transfusion rates that are similar to our cohort. A reduction in length of stay was not identified in the meta-analysis in contrast to our study.

This study has several limitations. Firstly, causal inference (internal validity) is limited by the non-randomised design and methods of analysis. However, this does represent a real-world description of the impact of IV iron therapy in an unselected cohort of CRC surgery patients in clinical practice. The generalisability (external validity) of our findings is uncertain due to the single-centre data collection, but the size of the cohort suggests that the findings are unlikely to be due to chance. As a real-world service, the patient population was limited to referrals to the POAS, and thus, a subset of the entire cohort of CRC patients was selected. Similarly, as the service expanded, more patients were treated, and fewer were referred who were not anaemic or not treated with iron. Furthermore, during the COVID-19 pandemic, the service was diminished both in terms of the number of patients treated and, importantly, data collection, as patients were operated on at other locations.

It was not possible to identify if patients with untreated anaemia received oral iron therapies. However, it has been shown that oral iron therapy in cancer patients has similar efficacy (improving Hb levels) to a placebo, so it may have had a limited impact on clinical outcomes [18]. Furthermore, any effect introduced by this limitation could reduce the effect size, and thus, our overall findings remain unchanged.

There was significant heterogeneity in the time between IV iron infusion and surgery in this population. This heterogeneity may, in part, have been caused by the COVID-19 pandemic, which delayed surgery in 2020 and 2021. It may also reflect geographical factors and limited opportunities to attend, as the UHS is a tertiary referral centre with a large catchment area. A minimum of four weeks from the time of infusion to surgery is optimal to achieve therapeutic benefit [8]. In a real-world setting, it may be unrealistic to achieve this in cancer pathways. However, there is evidence from emergency and smaller surgeries that IV iron given one day prior to or even on the day of surgery improves post-operative outcomes [19], and this may extend to post-operative IV iron infusions [20].

Despite using statistical techniques to control for confounding, given the non-randomised design of this study, we cannot exclude the possibility that the longer length of stay in the untreated anaemia group may have been caused by confounding factors that were unaccounted for. Most of these patients remained untreated because of late referral and insufficient time before surgery. This probably reflects the challenges of implementing a new referral pathway; in keeping with this, the untreated anaemia group reduced over time.

This study also has several strengths. It is an evaluation of the real-world implementation of an anaemia service and contributes to the evidence that the pre-operative correction of anaemia with IV iron is feasible within colorectal cancer pathways and improves outcomes after surgery. By reducing the rate of blood transfusion, IV iron may modify the reduced disease-free survival associated with iron deficiency anaemia [21] and alleviate the oncogenic risks associated with blood transfusion [22]. Furthermore, IV iron was associated with minimal adverse events in keeping with previous studies [8].

## 5. Conclusions

In a nested single-centre cohort study, pre-operative intravenous iron treatment for colorectal cancer patients with anaemia was associated with a reduced length of stay and reduced allogenic blood transfusion, but this was not associated with reduced readmission rates, unplanned ICU admissions, or reoperation rates.

## Figures and Tables

**Figure 1 cancers-17-01877-f001:**
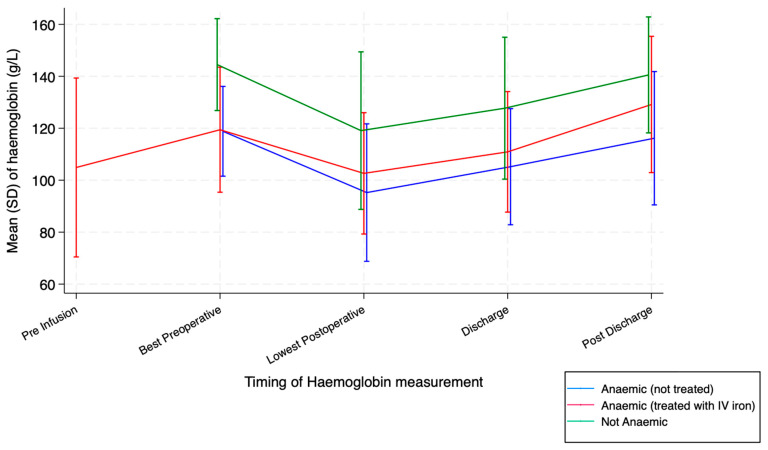
Comparison of mean haemoglobin levels across the perioperative period.

**Figure 2 cancers-17-01877-f002:**
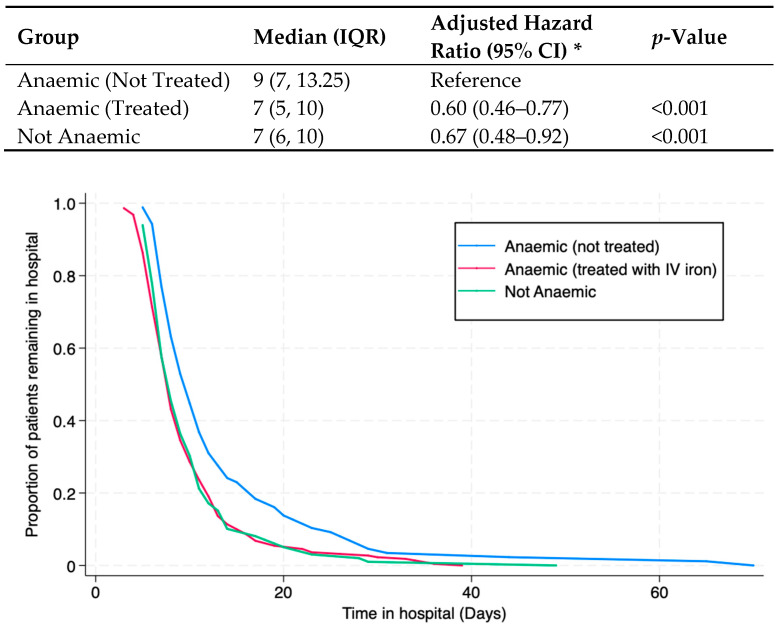
The median length of stay and the likelihood of remaining in hospital across anaemic treated, not treated and not anaemic patient groups. IQR = interquartile range; CI = confidence interval; * adjusted for age and sex.

**Table 1 cancers-17-01877-t001:** Baseline patient data across groups.

Characteristic		Not Anaemic (*n* = 99)	Anaemic, Treated (*n* = 220)	Anaemic, Not Treated (*n* = 88)	*p*-Value *
Patient Demographics					
	Age	65.24 (11.78)	71.02 (12.34)	72.35 (12.42)	0.393
	Sex (% Female)	29 (29)	106 (48)	54 (61)	0.036
	BMI	28.77 (4.67)	27.43 (5.76)	26.37 (5.94)	0.1537
Surgical Procedure					0.137
	Abdominal-perineal excision of rectum, *n* (%)	11 (11.1)	8 (3.6)	6 (6.8)	
	Anterior Resection, *n* (%)	42 (42.4)	54 (24.5)	32 (36.4)	
	Left Hemicolectomy, *n* (%)	5 (5.1)	14 (6.4)	3 (3.4)	
	Pan-Proctocolectomy, *n* (%)	1 (1.0)	2 (0.9)	1 (1.1)	
	Proctectomy, *n* (%)	1 (1.0)	0 (0)	1 (1.1)	
	Right Hemicolectomy, *n* (%)	36 (36.0)	135 (61.4)	43 (48.9)	
	Colectomy, *n* (%)	3 (3.0)	7 (3.2)	2 (2.3)	
Surgical Approach					0.544
	Laparoscopic, *n* (%)	75 (75.8)	147 (66.8)	60 (68.2)	
	Open Surgery, *n* (%)	24 (24.2)	70 (31.8)	28 (31.8)	
	Robotic Surgery, *n* (%)	0 (0)	3 (1.4)	0 (0)	
Comorbidities					
	Diabetes Mellitus, *n* (%)	12 (12.1)	46 (20.9)	12 (13.6)	0.140
	Hypertension, *n* (%)	34 (34.3)	93 (42.3)	44 (50)	0.218
	Ischaemic Heart Disease, *n* (%)	7 (7.1)	15 (6.8)	5 (5.7)	0.715
	Cerebrovascular Disease, *n* (%)	3 (3.0)	7 (3.2)	8 (9.1)	0.030
	COPD, *n* (%)	5 (5.1)	18 (8.2)	9 (10.2)	0.566

* *p*-value comparing patients with anaemia who were treated to those who did not receive treatment using a *t*-test for comparison of means or Chi^2^ tests.

**Table 2 cancers-17-01877-t002:** Pre- and post-operative blood test results for patients undergoing surgery for colorectal cancer.

	Anaemic (Treated)	Anaemic (Untreated)	Not Anaemic
Pre-infusion Hb (mean [SD], g/L)	104.90 (17.57)		
Pre-infusion TSAT (mean [SD], %)	12.43 (8.34)		
Pre-infusion ferritin (median [IQR], ng/mL)	14 (8, 30)		
Best Hb pre-operatively (mean [SD], g/L)	119.43 (12.28)	118.83 (8.82)	144.51 (9.02)
Lowest Hb post-operatively (mean [SD], g/L)	102.63 (11.91)	95.24 (13.50)	119.09 (15.47)
Hb at discharge (mean [SD], g/L)	110.91 (11.83)	105.20 (11.40)	127.72 (13.94)
Hb post-discharge (mean [SD], g/L)	129.15 (13.38)	116.16 (13.09)	140.53 (11.39)

**Table 3 cancers-17-01877-t003:** Progressively adjusted hazard ratios comparing time in hospital for patients who received IV iron treatment and patients who were not anaemic compared to patients with anaemia who did not receive treatment.

		Number of Patients	Hazard Ratio (95% Confidence Interval)	*p*-Value
Anaemia treated versus not treated				
	Unadjusted	308	0.63 (0.49 to 0.81)	<0.001
	Adjusted for age and sex	308	0.60 (0.46 to 0.77)	<0.001
	Adjusted for above and body mass index	301	0.60 (0.46 to 0.78)	<0.001
	Adjusted for above and surgical details	301	0.61 (0.47 to 0.80)	<0.001
	Adjusted for above and comorbidities	301	0.59 (0.45 to 0.78)	<0.001
Not anaemic versus non-treated anaemia				
	Unadjusted	187	0.64 (0.48 to 0.86)	0.003
	Adjusted for age and sex	187	0.67 (0.48 to 0.92)	0.015
	Adjusted for above and body mass index	184	0.69 (0.49 to 0.96)	0.027
	Adjusted for above and surgical details	184	0.66 (0.46 to 0.94)	0.021
	Adjusted for above and comorbidities	184	0.66 (0.45 to 0.95)	0.026

**Table 4 cancers-17-01877-t004:** Age- and sex-adjusted odds ratios comparing patients who received IV iron treatment and patients who were not anaemic compared to patients with anaemia who did not receive treatment for secondary outcomes.

Age- and Sex-Adjusted Models		Odds Ratio (95% Confidence Interval)	*p*-Value
Anaemia Treated versus Not Treated			
	Perioperative blood transfusion	0.35 (0.18 to 0.66)	0.001
	Unplanned admission to ICU	0.49 (0.16 to 1.46)	0.198
	Emergency return to theatre	0.37 (0.07 to 1.95)	0.243
	Unplanned readmission, 30 days	0.93 (0.23 to 3.75)	0.924
	Unplanned readmission, 8 weeks	0.94 (0.35 to 2.55)	0.903
Not Anaemic versus Not Treated			
	Perioperative blood transfusion	0.20 (0.07 to 0.55)	0.002
	Unplanned admission to ICU	0.22 (0.04 to 1.25)	0.087
	Emergency return to theatre	0.32 (0.04 to 2.28)	0.253
	Unplanned readmission, 30 days	1.35 (0.23 to 7.92)	0.738
	Unplanned readmission, 8 weeks	1.22 (0.33 to 4.49)	0.765

## Data Availability

The data and code that support the findings of this study are available from the corresponding author upon reasonable request.

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
