# Peer review of "Intravenous Iron for Perioperative Anaemia in Colorectal Cancer Surgery: A Nested Cohort Analysis"

_cancers, 2025, doi:10.3390/cancers17111877_

Round 1

Reviewer 1 Report

Comments and Suggestions for Authors

The present study evaluates the potential benefits of preoperative correction of iron deficiency anemia in patients undergoing surgery for colo-rectal cancer. The study is limited by its retrospective design, however it highlights the improvement in postoperative outcome. The study sample is large enough to support the statistical validity of the findings. The text is well written and organized, easy to follow, with tables and figures ensuring the information is presented in a reader-friendly manner. A few minor comments:

-on page 4, line 124, the authors specify the unit of henoglobin in g/dL, however the value of 130 corresponds to a unit of g/L (as it is also used throughout the rest of the text)

—the authors mention the heterogeneity of time intervals of iron supplementation prior to surgery as a potential limitation; while this is true, it would still be useful to specify in the results section the average number of days prior to surgery when the iron was administered, as well as the lenght of treatment (single dose or multiple administrations). 

Author Response

Dear Reviewer

Thank you for your comments. Please see below our specific responses

  1. on page 4, line 124, the authors specify the unit of haemoglobin in g/dL, however the value of 130 corresponds to a unit of g/L (as it is also used throughout the rest of the text)
  • Thank you, this is now corrected
  1. the authors mention the heterogeneity of time intervals of iron supplementation prior to surgery as a potential limitation; while this is true, it would still be useful to specify in the results section the average number of days prior to surgery when the iron was administered, as well as the length of treatment (single dose or multiple administrations). 
  • Treatment is generally with a single dose of IV iron, however multiple doses may be administered if needed. These are a minimum 4 weeks apart. I have added a description of this in the methods
  • Median (IQR) days to surgery from iron infusion is 25 (11, 41), I have added a sentence to this effect in the Results. (section 3.1 Treatment with intravenous iron)

Reviewer 2 Report

Comments and Suggestions for Authors

Authors of the manuscript investigated IDA, and the effect of preoperative IV iron in CRC. The study is well desribed, with a few minor issues to be resolved in methods and results. However, the discussion needs further extensions.

Major:

  1. Discussion is very short, basically omitting the comparison of the current study to previous ones. Please extend the discussion with such details.

Minor:

  1. Please consider adding an additional figure to show why patients were excluded from the whole cohort. Also please give some additional details, why the majority of patients had to be removed from the total of 2141 patients, resulting in 99+88+220.
  2. Fig1: Please indicate in the caption and in Methods as well, on which pre-/postoperative days the measurements were done (median / mean).
  3. Line 191: Authors wrote, n = 54 is 13%. The total investigated cohort was 220 + 99 + 88. Please, carefully look through the whole article, and check all numbers throughout the text.
  4. There are a few typos and double-spaces. Please, carefully look through the text.

Author Response

Dear Reviewer

Thank you for your comments. Please see below our specific responses.

Major:

  1. Discussion is very short, basically omitting the comparison of the current study to previous ones. Please extend the discussion with such details.
  • Thank you for your suggestion. We have expanded the discussion.

Minor:

  1. Please consider adding an additional figure to show why patients were excluded from the whole cohort. Also please give some additional details, why the majority of patients had to be removed from the total of 2141 patients, resulting in 99+88+220.

  • Thank you for this comment, and apologies if this was unclear. Patients were not excluded from the cohort. This was a pragmatic cohort of the subset of colorectal cancer patients undergoing elective surgery at UHS that were referred to the perioperative anaemia service.  This was a new service implemented in 2016, and not all patients undergoing colorectal surgery were referred. Furthermore between 2020 and 2022, referrals were disrupted by the COVID 19 pandemic. 

We have clarified the sentence in the methods to state:

  • We performed a nested cohort analysis of all elective CRC patients undergoing major surgical resections who were referred to the perioperative anaemia service (POAS) from 2016 to 2022. The anaemia service was disrupted during the COVID pandemic, reducing referrals over this period

We have also updated the results section to clarify this:

Over this period a total of 2141 patients underwent colorectal resections at UHS. This reflects the referral patterns from the surgical teams.  Non-anaemic patients are not routinely referred to the service

  1. Fig1: Please indicate in the caption and in Methods as well, on which pre-/postoperative days the measurements were done (median / mean).
  • Unfortunately, due to the way the data were collected, we are unable to do this within the timeframe specified. Haemoglobin levels are collected regularly throughout the perioperative period. We recorded
    • Hb pre-infusion
    • Highest Hb between infusion and surgery: as median time from infusion to surgery is 25 days, this value would be within this timeframe
    • Lowest Hb post-operatively: median length of stay was 7 days in the treated anaemia and not anaemic groups, and 9 days in the untreated anaemia group. Therefore, this value falls within this timeframe
    • Hb at discharge: This is taken on the date of discharge or day before if bloods weren’t measured on the day
    • Hb post-discharge: it is common for patients to return to clinic one to two weeks post discharge and thus blood tests were measured at this time. We included this to give us an idea of whether the Hb drop related to surgery improves over time.
    • I recognise that it would be interesting to map haemoglobin levels pre- and post-op in patients who did/ did not receive a blood transfusion. However, such work was not within the scope of this study, and we did not have sufficient granularity in our dataset to be able to undertake such a comparison.

 We have clarified these levels with the addition of a new table table 2 which gives mean/median levels at each time point

  1. Line 191: Authors wrote, n = 54 is 13%. The total investigated cohort was 220 + 99 + 88. Please, carefully look through the whole article, and check all numbers throughout the text.

  • Please can you clarify this comment? I have reviewed the numbers and 13% appears correct (54/407 = 0.1327. Happy to provide to one or two decimal places but we felt it did not add value in this setting.
  • There were 16 patients with missing transfusion data. While it did not change the numbers above, I have now hand-searched these data points from our EPR and updated the results section.
  • I have checked the analyses and updated as appropriate. There are no changes to our interpretation of our results.

  1. There are a few typos and double-spaces. Please, carefully look through the text.

  • Thank you – this has now been done, and we will do so again prior to publication

Reviewer 3 Report

Comments and Suggestions for Authors

The authors performed a secondary analysis of data prospectively collected in the Fit4Surgery Research Database. In this observational study, preoperative intravenous iron treatment was associated with reduced length of stay and blood transfusion. The topic analysed is interesting and not unequivocally evaluated. 

There are many strengths of the study:

  • 407 patients analysed
  • A proper statistical analysis with adjustment for potential confounders
  • Detailed description of baseline characteristics and methodology
  • Limitations of the study depicted properly in the discussion

I suggest minor corrections before publication of the manuscript:

1) In Table 1, the haemoglobin level in each group should be specified - mean value, minimum, and maximum - it is essential for interpreting the results.

2) Anaemia was defined as haemoglobin level less than 130 g/dl - it should be less than 13 g/dl.

3) No data on iron deficiency is included in Table 1. Mean ferritin level and TSAT should be mentioned. 

4) Anemia treatment was basen on Monofer iv according to national guidelines. A one sentence should be added that specifies whether it was administer once in all patients with treated anaemia.

5) The authors compare transfusion outcomes between groups. Transfusion may depend on intraoperative blood loss, not only on preoperative anaemia. Intraoperative blood loss should be reported in alaysed cohorts. 

6) Some English correction should be done: line 220 - should be 'causal inference' not 'casual inference'; line 201 'Wilcoxon' instead of 'Wilcoxen'

Author Response

Dear Reviewer, 

Thank you for your helpful, supportive comments on our paper. 

1) In Table 1, the haemoglobin level in each group should be specified - mean value, minimum, and maximum - it is essential for interpreting the results.

  • Thank you for your suggestion. We were not convinced overall mean Hb is useful, which is why we used Figure 1 showing Hb levels across the perioperative pathway. The numbers, for interest, are:
    • Best Hb pre-op: 125.66 (15.4), range 88-166
    • Lowest Hb Post-op: 105.19 (15.9), range 69 - 153
    • Hb at discharge: 113.93 (14.9), range 75 - 174
    • Hb post-discharge: 128.9 (15.4), range 82 – 170
  • Instead, I propose another table, showing mean (SD) Hb levels across the perioperative pathway as well as TSAT (mean, SD) and Ferritin (median, IQR) for patients treated with IV iron (as per your comment 3 below). This is now in the text as Table 2 and copied here:

Table 2. Pre- and post-operative blood test results for patients undergoing surgery for colorectal cancer

Anaemic (treated)

Anaemic

(untreated)

Not anaemic

Pre-infusion Hb (mean [SD], g/L)

104.90 (17.57)

Pre-infusion TSAT (mean [SD], %)

12.43 (8.34)

Pre-infusion Ferritin (median [IQR], ng/ml)

14 (8, 30)

Best Hb pre-operatively (mean [SD], g/L)

119.43 (12.28)

118.83 (8.82)

144.51 (9.02)

Lowest Hb post-operatively (mean [SD], g/L)

102.63 (11.91)

95.24 (13.50)

119.09 (15.47)

Hb at discharge (mean [SD], g/L)

110.91 (11.83)

105.20 (11.40)

127.72 (13.94)

Hb post-discharge (mean [SD], g/L)

129.15 (13.38)

116.16 (13.09)

140.53 (11.39)

2) Anaemia was defined as haemoglobin level less than 130 g/dl - it should be less than 13 g/dl.

- Thank you, this is now corrected. 

3) No data on iron deficiency is included in Table 1. Mean ferritin level and TSAT should be mentioned. 

- See above (comment 1). We only measured TSAT and Ferritin in patients with anaemia who received IV iron and I have presented these results. 

4) Anaemia treatment was based on Monofer iv according to national guidelines. A one sentence should be added that specifies whether it was administer once in all patients with treated anaemia.

 Now added, thank you

5) The authors compare transfusion outcomes between groups. Transfusion may depend on intraoperative blood loss, not only on preoperative anaemia. Intraoperative blood loss should be reported in alaysed cohorts. 

  • Unfortunately, we are unable to get intraoperative blood loss in the time frame specified. However, one of the reasons we analysed this particular group of patients, was because intraoperative blood loss across the CRC cohort is broadly similar compared to other surgical specialties.

6) Some English correction should be done: line 220 - should be 'causal inference' not 'casual inference'; line 201 'Wilcoxon' instead of 'Wilcoxen'

  • Thank you these have been corrected.

Round 2

Reviewer 2 Report

Comments and Suggestions for Authors

Authors answered all my questions, I have no further one(s).